# The Ontogeny and Dietary Differences in Queen and Worker Castes of Honey Bee (*Apis cerana cerana*)

**DOI:** 10.3390/insects15110855

**Published:** 2024-10-31

**Authors:** Chunyu Yang, Li Lei, Ying Wang, Baohua Xu, Zhenguo Liu

**Affiliations:** 1Key Laboratory of Efficient Utilization of Non-Grain Feed Resources (Co-Construction by Ministry and Province), Ministry of Agriculture and Rural Affairs, Tai’an 271018, China; 15064710371@163.com (C.Y.); leili_halcyon@163.com (L.L.); wangying@sdau.edu.cn (Y.W.); bhxu@sdau.edu.cn (B.X.); 2Shandong Provincial Key Laboratory of Animal Nutrition and Efficient Feeding, College of Animal Science and Technology, Shandong Agricultural University, Tai’an 271018, China

**Keywords:** *Apis cerana cerana*, life history, royal jelly, worker jelly, composition

## Abstract

The ontogeny of the queen and worker castes of bees is greatly affected by dietary differences. However, there are few studies that have addressed this issue in the honey bee *Apis cerana cerana*, a native subspecies of *Apis cerana*. Through an investigation into the nutritional differences between the diets of queen and worker larvae, as well as the changes in body weight during the growth and development processes of queens and workers, we found that the levels of protein and amino acids in the food of queen larvae are higher than those of worker larvae, and the rate of weight gain in queen larvae is also greater than that of worker larvae. These findings are valuable because the exploration of dietary differences can be helpful when carrying out conservation activities, especially in ex situ artificial rearing practices.

## 1. Introduction

The Chinese honey bee, *A. c. cerana*, is a subspecies of the *A. cerana* that evolved through long-term adaptation to China’s natural ecological environment [1]. This subspecies is distinctive within the country due to its notable attributes, including strong disease resistance to certain diseases, a wide foraging range, and resilience to both extreme cold and heat [2,3,4]. Thai sacbrood disease (TSBD) has previously erupted among *Apis cerana* in Thailand and India, resulting in a 100% mortality rate and causing significant devastation to the affected colonies [5]. However, there have been no reported cases of TSBD in *A. c. cerana*. *A. c. cerana* honey bees are prone to swarming, and this behavior is difficult to control. They may abscond for various reasons, such as a lack of food, the presence of ants, constant disturbances, or unsatisfactory locations [6]. Like the Western honey bee, both the queens and workers of *A. c. cerana* originate from fertilized eggs. However, workers are infertile females, whereas the queen is the sole fertile female within the colony, possessing well-developed ovaries [7,8]. The queen of *A. c. cerana*, akin to that of *A. m. ligustica*, experiences a 16-day developmental duration. However, *A. c. cerana* has smaller body dimensions in comparison to *A. m. ligustica*. Additionally, the workers of *A. c. cerana* possess a developmental timeline that is, on average, one day shorter than their Western honey bee counterparts, culminating in a total of 20 days [9]. Despite the well-illustrated ontogeny and dietary nutrition differences between the queen and worker castes of the honey bee *A. m. ligustica* [9,10], it is still unclear in the context of *A. c. cerana*.

The differentiation of queen and workers in *A. c. cerana* is simultaneously governed by nutritional status and the size of comb cells. Studies on *A. m. ligustica* have demonstrated that queen larvae exclusively consume royal jelly, while worker larvae ingest royal jelly during the first three days and then feed on a mixture of royal jelly, pollen, and honey [11,12]. A previous study suggested that the sugar content of royal jelly (RJ) during the first 1–3 days is approximately four times higher than that of worker jelly (WJ) [13]. Other studies have revealed that not only is the content of proteins, 10-HDA, glucose, and fructose significantly higher in RJ compared to WJ at the same age, but there are also significant differences in the moisture, proteins, minerals, and micronutrients between RJ and WJ across different ages [10,14]. Research on the nutritional regulation of caste differentiation in bees has primarily focused on *A. m. ligustica* [10,15,16], with limited studies addressing the differences in the nutritional composition of the RJ and WJ in *A. c. cerana*.

We speculate that the dietary differences are vitally important to the queen and worker cast differentiation in *A. c. cerana*. Accordingly, in this study, we (1) recorded the ontogeny duration, (2) measured the body weight, and (3) analyzed the dietary composition differences.

## 2. Materials and Methods

### 2.1. Bees

All the experimental investigations were conducted at the Experimental Apiary of Shandong Agricultural University (Tai’an, Shandong Province, China). The experiment utilized five colonies of Chinese honey bees (*A. c. cerana*) with equivalent colony strength. Production of RJ and WJ was collected in the five colonies at the same time. Each colony contained six frames, which included one sealed honey frame, half a frame of uncapped honey, half a frame of pollen, and four frames dedicated to brood rearing. The sealed brood occupied approximately one frame, and the bees could cover up to four frames. The colonies showed brisk foraging and reproduction activities and did not need supplementary feed. The RJ and WJ were collected from these five colonies on the same days from August to October 2023.

### 2.2. Developmental Stage of Workers

We tracked the individual developmental stage of the workers of *A. c. cerana* using the method described by Wang Ying [9]. To obtain larvae of the same age, we set up an empty comb in each beehive and kept it for 2 h to allow the workers to clean up any potential debris. An empty comb with one queen was placed into each frame cage and then placed in the center of the hive [17], allowing the queen to lay eggs on the empty comb for 12 h. Then, we placed the frame with eggs on it on the side of the hive away from the queen to prevent further egg laying. To reduce potential errors due to the variations in egg-laying time, we observed the hatching process every 5 h on the third day after egg laying. We covered the comb with a removable PVC transparent plastic sheet and marked the hatched larvae to precisely determine the developmental timeline of workers. We documented the complete developmental process of workers from egg to larva, then to pupa, and finally to emergence. For each developmental day, we randomly selected 30 samples for weighing to analyze the weight changes of *A. c. cerana* workers from egg to emergence. During the larval stage, we collected samples of WJ from 1–5-day-old specimens for nutritional analysis. This process was repeated multiple times due to the extremely limited availability of WJ in *A. c. cerana* and the substantial amount of WJ required for nutritional analysis.

### 2.3. Developmental Stage of Queens

We improved the previous rearing techniques used for *A. m. ligustica* queens [15,18]. A beaker filled with beeswax was placed in a water bath and heated to approximately 70 °C to ensure thorough melting of the beeswax. After we had immersed the queen cell molds in the liquefied beeswax, they were promptly submerged in cold water to allow the wax to solidify. Subsequently, the solidified queen cells were affixed to the queen-rearing frame (Figure 1). Each queen-rearing frame was affixed with 45 queen cells. One-day-old larvae were transferred into the artificially made queen cells and placed in the beehive for the workers to feed. On the 13th day after the larvae had been transferred, newly emerged queens could be obtained (which included 3 days of egg stages, totaling 16 days). The following day, we repeated the larval transfer process, enabling us to collect 15-day-old pupal-stage queens on the 12th day. By repeating this procedure, we were able to harvest queen samples at various developmental ages on the same day. Similarly, we also conducted image documentation for each developmental stage of the queen’s growth and development. We then randomly selected 30 queen samples at each developmental day for weighing and collected samples of 1- to 5-day-old RJ for nutritional analysis.

### 2.4. Measurement Methods

The moisture content of the RJ and WJ was determined using a vacuum freeze-drying method [19]. Following the weighing of the samples, they were placed in a freeze dryer (Christ, Osterode, Germany) and dried for 12 h. The difference in weight before and after drying constituted the water content. The nitrogen content, as determined by the Kjeldahl method, was used to calculate the crude protein content of the RJ and WJ [20]. The dried sample was introduced into a Kjeldahl digestion tube, followed by the addition of 0.4 g of anhydrous copper sulfate, 6 g of potassium sulfate, and 10 mL of concentrated sulfuric acid. The digestion tube was subsequently placed within a digestion furnace, where it was subjected to digestion for a period of 5 h. The crude protein content was thereafter quantified using an automated Kjeldahl nitrogen analyzer (Foss, Copenhagen, Denmark).

The amino acid content was detected via ultra-high performance liquid chromatography (UPLC Waters, Milford, MA, USA), as described by Hao and Luo [21]. Briefly, a Waters ACQUITY UPLC I-CLASS chromatography (Waters Corporation, Milford, MA, USA) equipped with a Waters UPLC HSS T3 column [2.1 mm (inner diameter) × 150 mm (length), 1.7 μm (particle dimension)] was used for separation with the column temperature at 50 °C. The mobile phase comprised a 0.1% aqueous solution of formic acid (Phase A) and acetonitrile (Phase B), with a flow rate set at 0.5 mL/min. The injection volume was 5.0 µL, and the elution gradient is detailed in Table 1. The MS data were recorded by a Waters XEVO TQ-S Micro system (AB SCIEX, Boston, MA, USA). The parameters were set as follows: ion source voltage at 1.5 kV, cone voltage at 20 V, desolvation temperature at 600 °C, desolvation gas flow at 1000 L/h, and cone gas flow at 10 L/h. The system was controlled by the Masslynx Analysis software (Masslynx V4.1 SCN 945, SCIEX, Boston, MA, USA).

### 2.5. Statistical Analysis

The recorded data were statistically analyzed with one-way ANOVA using SPSS Statistical software (IBM SPSS Statistics 18 software, Chicago, IL, USA). Tukey’s HSD was used as a post hoc test. Accordingly, multiple comparisons of the moisture and crude protein contents between the RJ/WJ groups of different ages were performed. Independent sample *t*-tests were used for pairwise comparisons of the moisture, crude protein, and amino acid contents between the RJ and WJ of the same age. The treatment means were compared at a 5% level of significance.

## 3. Results

### 3.1. The Larval and Pupal Morphological Characteristics of A. c. cerana Queens and Workers

The development of the queen included a 3-day egg stage, a 6-day larval stage, and a 7-day pupal stage. The developmental process is illustrated in Figure 2. The queen eggs hatched into larvae on the 4th day, and these eggs were relatively small and encircled by transparent RJ. By the 5th day, each larva expanded to occupy nearly half of the cell’s bottom, with an increased volume of RJ surrounding it. As development progressed, the larva grew rapidly; by the 6th day, it filled most of the cell’s base, and the RJ around it became light yellow. On the 7th day, the larva occupied the entire cell base, with a significant amount of food being visible around it. By the 8th day, the larva filled the entire cell, and the remaining food at the bottom became sparse and viscous. At this stage, the queen cell had been capped. On the 9th day, the larva consumed all the remaining food and began adjusting its body position in preparation for pupation. The following day, the larva entered the prepupal stage, adopting a head-down posture and ceasing to feed until emergence. Between the 11th and 13th days, the larva metamorphosed into a white pupa, followed by the eye color changing from white to pink and then to black. By the 14th day, melanin accumulation caused the pupa’s head and thorax to turn brown. On the 15th day, the body color of the pupa darkened in preparation for the queen’s emergence. The queen emerged on the 16th day.

Compared to the queens, the developmental time for *A. c. cerana* workers was longer, where the pupal stage of workers alone lasted 11 days and the full developmental stage took a total of 20 days. The developmental stages are depicted in Figure 3. The initial 9 days of worker development closely resembled those of the queens, encompassing a 3-day egg stage followed by a 6-day larval stage. On the 9th day, the worker cells were capped. From the 10th to 12th day, the workers transitioned into a 3-day prepupal stage, during which the pupa ceased to feed, adjusted its head to a fixed orientation, and, afterward, remained immobile. Between the 13th and 16th days, the pupae’s body and head remained white, with progressive color changes occurring in the eyes (from white to pink, then to brown, and finally to black), allowing us to estimate the developmental age by eye color. On the 17th day, the pupae’s head and thorax exhibited a yellowish hue due to the accumulation of pigments. On the 18th day, this coloration intensified, resulting in a light black appearance of the pupae’s body. On the 19th day, the volume and weight of the pupae decreased in preparation for emergence. On the 20th day, the newly emerged workers exited the cell and began to eat food.

### 3.2. The Average Body Weight of the Queen and Workers During Ontogeny

Table 2 shows the changes in weight for *A. c. cerana* workers and queens throughout their developmental process. The average weight of both queen and worker eggs was 0.18 mg. On the 4th day, the eggs hatched into larvae, and the average weight of both queen and worker larvae was 0.54. The daily weight gain of the queen and worker bee larvae both exhibited a trend of initial increase followed by decrease. The peak weight gain percentage for the queen larvae occurred on the 6th day, reaching 5.86%; for the worker bee larvae, the peak weight gain percentage was observed on the 7th day, amounting to 4.37%. The weight of both queen and worker larvae peaked on the 9th day, with average weights of 230.69 mg and 126.80 mg, respectively, and the maximum weight of a queen larva was approximately twice that of a worker larva. After this point, the weights of both the queen and worker larvae began to decrease gradually. On the 16th day, when the queen emerged from the queen cell, the average weight of a 1-day-old queen was 154.75 mg. On the 20th day, when the worker emerged from the worker cell, the average weight of a 1-day-old worker was 84.5 mg.

### 3.3. Changes in Moisture and Crude Protein Contents in RJ and WJ at Different Ages

Appendix A shows the variation in the moisture and crude protein contents of the RJ and WJ at different ages. The moisture content in the WJ exhibited a trend of initially increasing and then decreasing, with the highest moisture content of 76.55% observed in 3-day-old WJ. By contrast, the moisture content in the 5-day-old WJ was only 58.54%, which was significantly lower than at other ages (*p* < 0.05). The moisture content in the 2- to 5-day-old RJ also showed a trend of increasing initially and then decreasing. The moisture content in the 2-day-old and 5-day-old RJ was very similar at 57.51% and 58.09%, respectively, and it was significantly lower than in other ages (*p* < 0.05).

The crude protein content in the WJ exhibited a trend of initially increasing and then decreasing, with the highest protein content of 54.85% observed in the 3-day-old WJ. By 5 days of age, the protein content in the WJ dropped to 26.56%, which was significantly lower than that at other ages (*p* < 0.05). There was no significant difference in the crude protein content between the 2-day-old and 3-day-old RJ, with the highest protein content of 51.97% found in the 3-day-old RJ. By Day 5, the protein content in RJ decreased to 40.41%, which was significantly lower than that at other ages (*p* < 0.05).

A comparison of the moisture and crude protein contents between WJ and RJ of the same age revealed that, as shown in Figure 4, the moisture content in the 2-day-old WJ was significantly higher than that in the 2-day-old RJ (*p* < 0.001), and the moisture content in the 3-day-old WJ was significantly higher than in the 2-day-old RJ (*p* < 0.01). Additionally, the crude protein content in the 5-day-old WJ was significantly lower than in the 5-day-old RJ (*p* < 0.001), while differences in the crude protein content between the WJ and RJ at other ages were not significant.

### 3.4. Changes in Amino Acid Content in RJ and WJ at Different Ages

The results of amino acid analysis for the WJ and RJ at different ages are shown in Table 3. The amino acid content in the WJ from the first three days of age was relatively consistent, but there was a noticeable decline in the amino acid levels in the WJ from days 4 and 5, and it was particularly noticeable on day 5, when most amino acids were significantly lower compared to those in the first three days. As the age of the WJ increased, the methionine content gradually decreased. The methionine content in the 1-day-old WJ was significantly higher than that in the 3–5-day-old WJ samples. The cystine content did not show significant differences across different ages of WJ, but it generally followed a trend of increasing and then decreasing, with its highest level observed on day 3. Aside from methionine and cystine, the levels of other amino acids were highest in the 1-day-old WJ, followed by the 3-day-old WJ, with lower levels observed in the 4- and 5-day-old WJ.

The amino acid content in the RJ samples at different ages generally showed a trend of increasing and then decreasing. The amino acid levels (excluding cystine and methionine) were highest in the 3-day-old RJ compared to at other ages. The cystine content was relatively low in the RJ samples, with minimal variation, and it peaked on day 4. The methionine content of the 1-day-old RJ sample was the highest, which was significantly higher than that of the 2-, 4-, and 5-day-old RJ samples.

In comparing the amino acid content between the same aged RJ and WJ, we found that most amino acid levels were significantly higher in the RJ compared to the WJ. In the 4-day-old RJ, the histidine content was significantly higher than in the WJ, while the histidine content in the RJ and WJ at other ages did not show significant differences. The cystine content was lower in both the RJ and WJ, which was why there was little difference between the two. The methionine levels were similar in the RJ and WJ samples; however, at 2 days of age, the methionine content in the RJ was less than in the WJ, which was the only case where an amino acid was less abundant in the RJ compared to the WJ.

## 4. Discussion

Currently, extensive research has been conducted on the morphological and developmental characteristics of Africanized honey bees (a hybrid of the Western honey bee *A. mellifera* and African honey bee *A. m. scutellate*) and *A. m. ligustica* [9,22]. This study clearly illustrates the developmental process of *A. c. cerana* queens within queen cells and workers within worker cells in a regular honeycomb, as well as the changes in body weight during the growth of queens and workers.

We previously compared the developmental durations and body weight differences of newly emerged bees among several species [9,22,23,24]. Our findings reveal that the developmental process of *A. c. cerana* workers bears similarity to that of Africanized honey bees. The developmental durations of these two species are comparable, averaging approximately 20 days, with the average weight of newly emerged workers differing by only 0.1 mg. Furthermore, the developmental duration of *A. c. cerana* queens is analogous to that of *A. m. ligustica*, averaging 16 days. However, a notable disparity was observed in the newly emerged *A. c. cerana* queens, which were lower compared to *A. m. ligustica* and Africanized honey bees. Given that *A. c. cerana* requires nesting in environments with varied topography, climate, and nectar plants, a high resilience to changes in the external environment is also exhibited [25]. The differences between *A. c. cerana* and other bee species may be precisely the adaptations they have made to accommodate the environment [1]. Studies have shown that both queen and worker larvae attain their maximum body weight on the 9th day of development, with the queens exhibiting greater pupal weights, faster growth rates, and shorter developmental durations compared to the workers [9,22,26]. In the present study, it was observed that both the queen and worker larvae of *A. c. cerana* reached their peak body weight on the 9th day. Additionally, the weight of the 9th-day queen larvae was approximately twice that of the worker larvae, a finding that aligns with previous research. Studies have shown that there is a significant correlation between the size of the brood cells and the morphology of the bees in *A. c. cerana* [27]. It is the queen cell size that is notably larger than that of the worker cells, which thus constitutes a significant factor leading to the substantial disparity in body weight between queen and worker larvae on the 9th day.

It is currently widely accepted that worker larvae exclusively consume RJ during the first three days of their development and then subsequently transition to a diet comprising a mixture of RJ, honey, and pollen [11,12]. However, our findings indicate that the moisture content of WJ on days 2 and 3 is significantly higher than that of RJ. Moreover, we observed the presence of small quantities of pollen grains in the collected larval food. These observations lead us to hypothesize that the larvae of *A. c. cerana* may begin consuming a mixture of RJ, honey, and pollen as early as the second day.

Amino acids are the basic units of proteins; as such, changes in the total amino acid content can reflect variations in protein levels. Research has indicated that the protein and amino acid contents in RJ follow the same trend over time [28]. In our research, the protein content in the RJ initially increased and then decreased, peaking at day 3, and the amino acid content showed a similar trend, which is consistent with previous findings [28]. However, our study also found that WJ did not exhibit this pattern. Most amino acids in WJ showed a trend of decrease–increase–decrease, which does not align with the protein content trend (increase then decrease). This discrepancy might be due to the fact that we measured the protein content using dried WJ, while the amino acids were assessed in fresh samples. Additionally, the increased moisture content in WJ on day 2 might have diluted the samples, thus influencing the results.

We identified the five most abundant amino acids on day 3 RJ and WJ as Asp, Glu, Leu, Lys, and Val. This result is consistent with previous studies [28]. However, there is a slight difference when compared to Zhao’s findings [29], which identified Asp, Glu, Pro, Leu, and Lys as the top five amino acids. In honey bees, a significant disparity exists between the lifespan of workers and queens, with workers surviving for mere weeks, whereas queens can persist for years [30]. Research has suggested that restricting methionine can extend the lifespan of rodents and fruit flies [31,32,33,34]. In our study, the concentrations of most amino acids in royal jelly (RJ) were found to be higher than those in worker jelly (WJ) of the same age, with the exception of methionine. Notably, the methionine levels in RJ and WJ on days 3 and 4 were comparable, and WJ on day 2 exhibited a higher methionine content than RJ. Therefore, the difference in methionine content between RJ and WJ may be an important factor for the difference in longevity between queens and workers.

## 5. Conclusions

Previous studies on *A. m. ligustica* and the Africanized honey bee have provided a detailed ontogenesis of queens and workers [9,22,24,35,36]. These descriptions are extremely valuable for scientific studies of honey bees and other bee pollinators. To the best of our knowledge, this is the first detailed study of the individual developmental processes of *A. c. cerana* queens and workers. It includes an analysis of nutritional components in the RJ and WJ for the first five days and provides an in-depth investigation into the relationship between the nutritional contents of these two types of bee larval food. The results offer theoretical support for ex situ artificial rearing practices of *A. c. cerana*.

## Figures and Tables

**Figure 1 insects-15-00855-f001:**
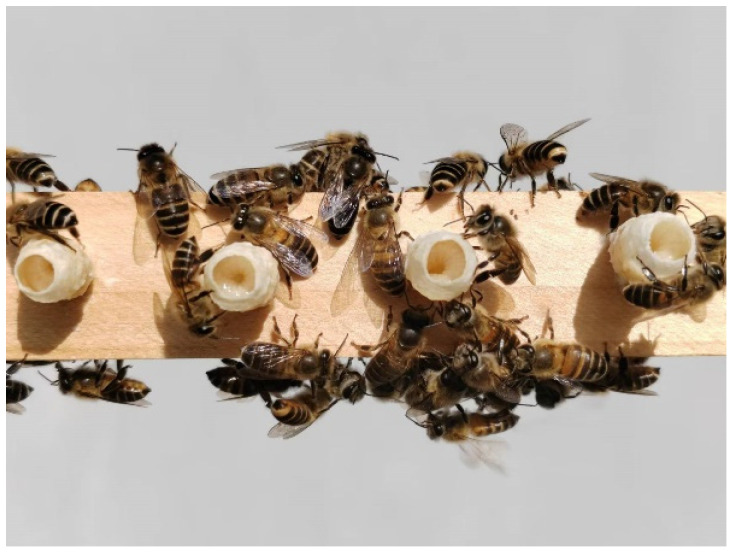
The queen-rearing process of *A. c. cerana*.

**Figure 2 insects-15-00855-f002:**
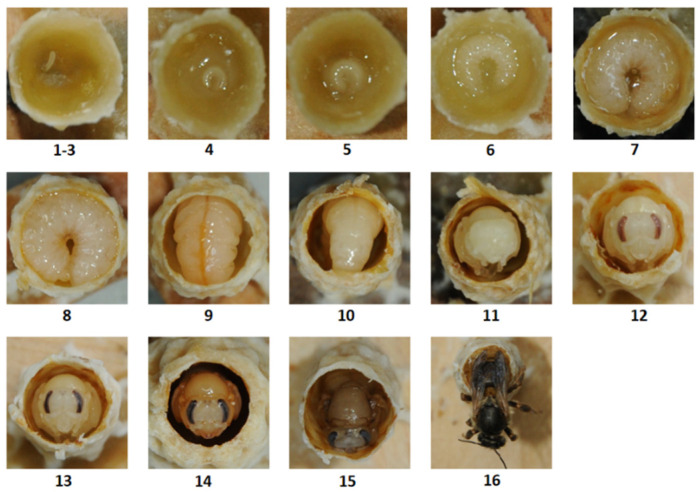
An *A. c. cerana* queen in a queen cell at different stages of development. The numbers indicate the developmental days after egg laying.

**Figure 3 insects-15-00855-f003:**
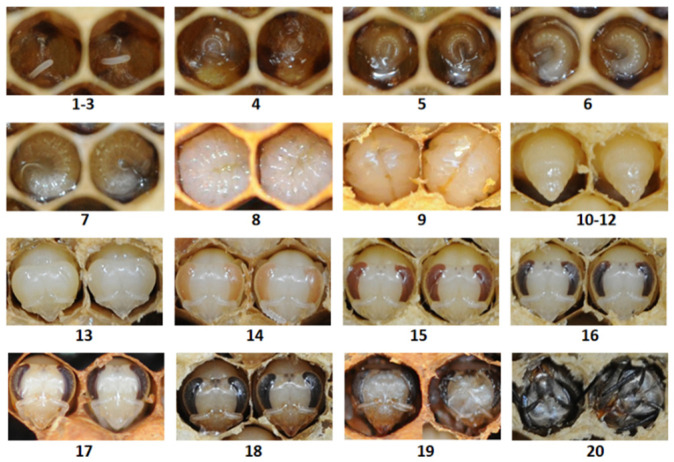
*A. c. cerana* workers in the worker cells at different stages of development. The numbers indicate the developmental days after egg laying.

**Figure 4 insects-15-00855-f004:**
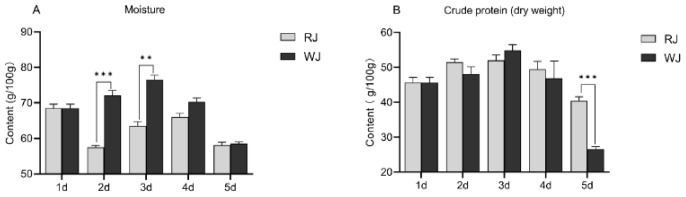
Variation in the moisture (**A**) and crude protein (**B**) contents in the WJ and RJ of *Apis cerana cerana* at different ages. ** *p* < 0.01, *** *p* < 0.001.

**Table 1 insects-15-00855-t001:** Elution gradients used for amino acid analysis.

Time (min)	Mobile Phase	
	A (v%)	B (v%)
0	96	4
0.5	96	4
2.5	90	10
5	72	28
6	5	95
7	5	95
7.1	96	4
9	96	4

Note: Mobile Phase A: water and 0.1% formic acid solution; Mobile Phase B: acetonitrile.

**Table 2 insects-15-00855-t002:** Body weight of the *A. c. cerana* queens and workers during ontogenetic development. The data are reported as the means ± SE (*n* = 30).

Day of Development	Body Weight (mg)	
	Queen	Worker
1–3	0.18 ± 0.01	0.18 ± 0.01
4	0.54 ± 0.02	0.54 ± 0.02
5	2.28 ± 0.06	2.15 ± 0.05
6	15.64 ± 0.36	6.54 ± 0.22
7	69.58 ± 0.64	35.12 ± 0.71
8	180.66 ± 3.33	85.67 ± 1.18
9	230.69 ± 1.95	126.80 ± 0.70
10	216.03 ± 2.24	112.70 ± 0.73
11	208.30 ± 1.87	109.39 ± 0.82
12	202.50 ± 1.97	105.85 ± 1.09
13	195.26 ± 2.07	100.77 ± 0.69
14	181.69 ± 1.65	99.92 ± 0.67
15	172.03 ± 1.86	97.28 ± 0.78
16	154.75 ± 2.65	96.10 ± 0.83
17	—	95.85 ± 0.73
18	—	92.28 ± 0.42
19	—	89.98 ± 0.52
20	—	84.50 ± 0.63

**Table 3 insects-15-00855-t003:** Amino acid content of the RJ and WJ of *A. c. cerana* (%, fresh weight) on different days. Data are reported as the mean ± SE (*n* = 3). An asterisk (*) next to a higher value of the RJ or WJ indicates a significant difference between the RJ and WJ samples on the corresponding day at *p* < 0.05. Different lowercase letters (a, b, and c) in a column indicate significant differences among the RJ or WJ samples at *p* < 0.05.

Amino Acid Content %		1 d	2 d	3 d	4 d	5 d
Histidine (His)	WJ	0.35 ± 0.04 a	0.34 ± 0.03 a	0.35 ± 0.01 a	0.31 ± 0.01 ab	0.24 ± 0.01 b
	RJ	0.35 ± 0.04	0.42 ± 0.04	0.49 ± 0.05	0.46 ± 0.03 *	0.36 ± 0.04
Arginine (Arg)	WJ	0.77 ± 0.08 a	0.63 ± 0.02 ab	0.65 ± 0.03 ab	0.49 ± 0.02 c	0.55 ± 0.01 bc
	RJ	0.77 ± 0.08 bc	0.94 ± 0.07 *ab	0.97 ± 0.03 *a	0.78 ± 0.04 *bc	0.76 ± 0.02 *c
Serine (Ser)	WJ	0.45 ± 0.04 a	0.39 ± 0.01 ab	0.43 ± 0.02 a	0.38 ± 0.02 ab	0.35 ± 0.01 b
	RJ	0.45 ± 0.04 b	0.57 ± 0.03 *a	0.59 ± 0.01 *a	0.56 ± 0.01 *a	0.53 ± 0.02 *ab
Glycine (Gly)	WJ	0.49 ± 0.05 a	0.42 ± 0.01 ab	0.46 ± 0.02 a	0.41 ± 0.03 ab	0.37 ± 0.01 b
	RJ	0.49 ± 0.05 b	0.59 ± 0.03 *ab	0.62 ± 0.01 *a	0.57 ± 0.02 *ab	0.54 ± 0.03 *ab
Aspartic acid (Asp)	WJ	1.72 ± 0.18 a	1.55 ± 0.01 ab	1.67 ± 0.02 ab	1.37 ± 0.08 bc	1.24 ± 0.04 c
	RJ	1.72 ± 0.18 b	2.32 ± 0.16 *a	2.42 ± 0.04 *a	2.21 ± 0.11 *a	2.08 ± 0.13 *ab
Glutamic acid (Glu)	WJ	1.29 ± 0.14 a	1.15 ± 0.01 ab	1.27 ± 0.01 a	1.07 ± 0.05 ab	0.98 ± 0.02 b
	RJ	1.29 ± 0.14 b	1.69 ± 0.11 *a	1.79 ± 0.05 *a	1.60 ± 0.04 *ab	1.49 ± 0.10 *ab
Threonine (Thr)	WJ	0.61 ± 0.06 a	0.52 ± 0.01 ab	0.58 ± 0.02 a	0.51 ± 0.03 ab	0.46 ± 0.01 b
	RJ	0.61 ± 0.06 b	0.76 ± 0.04 *a	0.80 ± 0.02 *a	0.74 ± 0.01 *a	0.70 ± 0.03 *ab
Alanine (Ala)	WJ	0.52 ± 0.05 a	0.45 ± 0.01 abc	0.48 ± 0.02 ab	0.42 ± 0.02 bc	0.38 ± 0.01 c
	RJ	0.52 ± 0.05 b	0.63 ± 0.04 *ab	0.67 ± 0.02 *a	0.62 ± 0.01 *ab	0.59 ± 0.03 *ab
Proline (Pro)	WJ	0.46 ± 0.04 a	0.39 ± 0.01 ab	0.44 ± 0.02 ab	0.39 ± 0.02 ab	0.38 ± 0.01 b
	RJ	0.46 ± 0.04 b	0.56 ± 0.03 *a	0.59 ± 0.02 *a	0.54 ± 0.01 *ab	0.52 ± 0.03 *ab
Lysine (Lys)	WJ	0.99 ± 0.09 a	0.85 ± 0.01 ab	0.93 ± 0.04 ab	0.81 ± 0.04 bc	0.68 ± 0.01 c
	RJ	0.99 ± 0.09 c	1.31 ± 0.07 *ab	1.37 ± 0.03 *a	1.25 ± 0.05 *ab	1.12 ± 0.05 *bc
Cysteine (Cys)	WJ	0.07 ± 0.02	0.07 ± 0.01	0.08 ± 0.01	0.06 ± 0.01	0.04 ± 0.01
	RJ	0.07 ± 0.02	0.10 ± 0.02	0.10 ± 0.01	0.11 ± 0.02	0.08 ± 0.01
Tyrosine (Tyr)	WJ	0.55 ± 0.07 a	0.46 ± 0.01 ab	0.47 ± 0.02 a	0.36 ± 0.01 bc	0.34 ± 0.01 c
	RJ	0.55 ± 0.07 b	0.65 ± 0.04 *ab	0.70 ± 0.01 *a	0.59 ± 0.03 *ab	0.54 ± 0.02 *b
Methionine (Met)	WJ	0.29 ± 0.04 a	0.21 ± 0.02 ab	0.15 ± 0.04 b	0.11 ± 0.03 b	0.10 ± 0.03 b
	RJ	0.29 ± 0.04 a	0.12 ± 0.04 b	0.16 ± 0.03 ab	0.11 ± 0.03 b	0.15 ± 0.04 b
Valine (Val)	WJ	0.87 ± 0.07 a	0.73 ± 0.01 b	0.77 ± 0.02 ab	0.64 ± 0.04 bc	0.58 ± 0.01 c
	RJ	0.87 ± 0.07 b	1.02 ± 0.06 *ab	1.10 ± 0.03 *a	1.00 ± 0.03 *ab	0.93 ± 0.05 *ab
Isoleucine (Ile)	WJ	0.75 ± 0.07 a	0.64 ± 0.01 ab	0.67 ± 0.02 ab	0.57 ± 0.03 bc	0.50 ± 0.01 c
	RJ	0.75 ± 0.07 b	0.89 ± 0.06 *ab	0.95 ± 0.03 *a	0.87 ± 0.02 *ab	0.81 ± 0.04 *ab
Leucine (Leu)	WJ	1.14 ± 0.11 a	0.98 ± 0.01 ab	1.05 ± 0.04 ab	0.87 ± 0.05 bc	0.77 ± 0.01 c
	RJ	1.14 ± 0.11 b	1.40 ± 0.08 *a	1.52 ± 0.04 *a	1.38 ± 0.04 *a	1.27 ± 0.07 *ab
Phenylalanine (Phe)	WJ	0.64 ± 0.06 a	0.53 ± 0.01 bc	0.57 ± 0.02 ab	0.48 ± 0.02 bc	0.43 ± 0.01 c
	RJ	0.64 ± 0.06 c	0.78 ± 0.05 *ab	0.84 ± 0.02 *a	0.76 ± 0.02 *abc	0.70 ± 0.03 *bc
Total Amino Acid (TAA)	WJ	11.95 ± 1.19 a	10.30 ± 0.12 ab	11.03 ± 0.34 ab	9.26 ± 0.47 ab	8.83 ± 0.14 b
	RJ	11.95 ± 1.19 b	14.76 ± 0.94 *ab	15.70 ± 0.32 *a	14.19 ± 0.51 *ab	13.17 ± 0.66 *ab

## Data Availability

The original contributions presented in this study are included in the article; further inquiries can be directed to the corresponding authors.

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
