# Peer review of "The Ontogeny and Dietary Differences in Queen and Worker Castes of Honey Bee (Apis cerana cerana)"

_insects, 2024, doi:10.3390/insects15110855_

Round 1

Reviewer 1 Report

Comments and Suggestions for Authors

The Asiatic hive bee, Apis cerana has been  less studied. I will  encourage the authors to keep working on the biology of this honey bee. This is a good article on the endemic honey bee species of China, Apis cerana cerana. This article carries some useful information on the Chinese honey bee, Apis cerana cerana which are new to Science and worth publication in Insects. However, English language needs special attention. The article should be read by an English language expert. The article in its present form has some lacunae which need to be rectified. I, therefore, recommend a major revision of this article.

Specific comments

Title:

Title should read “The  ontogeny and dietary differences in queen and worker castes of honey bee, Apis cerana cerana

Abstract:

This needs rewriting. In the first 2-3 lines write the purpose of this study (e. g. Apis cerana cerana is endemic in China, This species has been very less studied. For its efficient utilization in beekeeping and pollination, there is a need to study its biology and management practices. With this background the present study was carried out). Then in 3-4 lines write briefly the Methods followed to conduct this study. Then in 4-5 lines write the salient Results of this study. In the last, write in 3-4 lines conclusion and projections of this study.

Line 11: Give authority of Apis cerana cerana

Introduction:

Again, this Section too needs modification. In the first paragraph write the purpose of this study. In the second/and third paragraph review the relevant literature. In the last paragraph build your hypothesis and clearly define the objectives of this study. The results mentioned in the last paragraph of present Introduction is redundant and out of context. This may be deleted.

Line 23: strong disease resistance—Apis cerana had Thai Sac Brood Disease in some Asian countries; starting from Thailand through India. What was the status of this disease in China? Make a mention of this, and if this disease was absent in China, then write about this positive trait in A.c.cerana.

Line 24: resilience to both extreme cold and heat—Apis cerana absconds due to small temperature changes in south and South-East Asia. If this is absent in A.c. cerana, make a mention of this positive trait too in A.c.cerana.

Material and Methods

2.1. Bees: In this section, explain what types of colonies were used? How much unsealed brood, sealed brood, pollen and honey stores and colony strength were there? You may refer: doi: http://dx.doi.org/10.5897/JENE2017.0679

Lines 54-55: apiculture experimental station-Apiculture Experimental Station

Lines 64, 66 and elsewhere: What is an egg laying controller? I have not heard about this terminology.

Lines 61-71: This part is confusing. Can you make these contents understandable by some drawings? Or, you rephrase these sentences to make them clear and well understandable. Kindly write the words like empty comb, brood comb etc.

Line 71: Which queen and workers? Adult or developing?

Line 75: 2.3. Breeding of the queen bee- This is not queen breeding, this is queen rearing; replace the word elsewhere too.

Lines 79-82: write complete sentences in the past tense.

Lines 86, 89: Moisture content of which material? Write here.

Table 1: What is elution gradient; what are A and B? Write in the foot note.

Statistical Analysis

First give a layout plan of your experiment e.g. Number of Treatments (what they are?), number of replications (n=?). Then you justify your Statistical design used for your recorded data. What for did you use the independent t-test? Elaborate.

This section can be written in simple language, e.g.

“The recorded data were statistically analysed in one-way ANOVA using SPSS Statistical software. Tukey’s HSD was used as post hoc test. Accordingly, multiple comparisons of moisture and crude protein content between different age RJ/WJ groups were done. Independent samples t-test was used for pairwise comparison of moisture, crude protein, and amino acid content between RJ and WJ of the same age. The treatment means were compared at 5% level of significance”

Table 2. Heading: write the number of observations n e.g. means ± SE. (n=? )

Figure 4: Increase the size of this figure to depict the words more clear.

Table 3: Heading: write the number of observations n e.g. means ± SE. (n=? )

Lines 273-275: Not clear: kindly rephrase the sentences.

Lines 273-281: Give this paragraph a sub-head, Conclusion.

Add sub-head ‘Acknowledgements’ and write the physical facilities used from the Department/Laboratory, and any other person, Head of the Department or Laboratory pesonels etc., if any.

References: By and large ok.

Comments on the Quality of English Language

language at some places is confusing, and needs rectification. 

Reviewer 2 Report

Comments and Suggestions for Authors

General comments

This manuscript contributes to valuable basic biological information for eastern honey bee development. It will improve our understanding in honey bee nutrition and the differentiation of castes in honey bees. The grammatical errors and language should be revised and improved throughout the manuscript before publication. The authors are suggested to use past tense for describe their research methods and findings. A few method details should be added, such as numbers of subjects (eggs, larvae and pupae) and rearing frame used for this study (refer to the detailed comments). Although many grammatical errors have been mentioned, the errors of the same type should be corrected throughout the manuscript.

Detailed comments

Line 2: the manuscript does not demonstrate diet diversity instead diet nutrition variations. Authors studied developmental stage not the whole life of honey bees. Suggest to change the title, for example, “The growth and development and diet nutrition of queen and worker bees, Apis cerana cerana”.

Line 11, replace “possesses” with “possessing”.

Line 24, add “Like western honey bee”

Line 26, deleted “temporarily”.

Line 28, replace “Nevertheless” with “however”.

Line 29, replace “bodily” with “body”

Line 31, replace “Western” with “western honey bee”.

Line 33, replace similarly with “simultaneously”.

Line 34, replace “factors” with “status” or “condition”.

Line 36, delete “subsequently”.

Line 37, replace “early study” with “A previous study”.

Line 37-38, is this sentence right? In the following sentence, authors mentioned royal jelly has higher sugar (glucose and fructose) content.

Line38-39, replace recent research with “Other studies have”.

Line 41, replace “trace elements” with “micronutrients”.

Line 42, delete “Currently”.

Line 46, delete “comprehensive”.

Line 49, clarify “at various day ages”.

Line 50, delete “ancient and”. Authors do not study ancient bees, and both Apis mellifera and Apis cerana have their ancient ancestors. Suggest to mention other potential implications of this study, such as “This study will provide insights into understanding of nutritional requirement of A. cerana of worker and queen bees throughout of their development, particularly protein and amino acid nutrition demand.”

 Lines 54-55, change “apiculture experimental station” to “experimental apiary”.

Line 56, delete “production”.

Line 57, change “in the” to “from these”. Combine two sentences “RJ” and “WJ…” and “Sampling of ….” The suggested new sentence is “RJ and WJ production 56 were collected from these five colonies on the same days from August to October in 2023.”

Line 60, change “developmental process” to “developmental stage”.

Line 61, did you introduce one empty frame in each of five hives?

Line 61, change “To accurately control the age of the 61 larvae” to “To gain larvae of the same age,” Combine the sentences in line 61-62, for example “To gain larvae of the same age, we set up an empty comb in each bee hive and kept them for xxx days to allow worker bees to clean up potential debris.)”

Line 64, rewrite “the queen and the new comb” to “one empty comb with one queen”.

Line 65, replace “forcing” with “allowing” and “the hive” with “a hive”

Line 72, from five hives? Delete “starting”.

Line 73-74, describe how many samples (queens and workers) used for this study.

Line 75, replace “Breeding” with “Rearing”. Authors are not breeding instead rearing.

Line 76, suggest to add “by coating plastic queen rearing cells with bee wax” in the end of the sentence.

Line 77-78, rewrite the sentence. The suggestions are “We melted A. c. cerana wax at high temperature and dipped queen cell molds in hot liquid wax. The queen cell molds cooled down in cold water and attached to a queen-rearing frame (Fig. 1).” Please estimate the potential temperature you used for your study. Is high temperature at 50 or 100 Celsius degree, or other degrees?

Line 79, please describe how many cells authors installed on each rearing frame. How many frames were used? How many frames did each hive receive?

Line 79, suggest changes “One-day old larvae were transferred into queen rearing cells on each rearing frame, and then moved to each hive.”

Line 80, delete “systematically”

Line 80, what does “by adjusting the collection schedule” mean?

Line 85, replace “Detection method” with “Nutrition analysis method” or “Measurement methods”

Line 93, replace “of 50°C” with “at 50°C”.

Line 108, suggest to change “Elution gradient” to “Elution gradients used for amino acid analysis.”

Line 111, delete “for variance analysis”. ANOVA means “analysis of variance”.

Line 112, suggest to revise the sentence to “comparisons of moisture and crude protein content in RJ and WJ between different ages.”

Line 115, suggest to replace “the comparison groups” with “ages or jelly types”

Line 118, suggest change the subtitle to “The larval and pupal morphological characteristics of A. c. cerana queens and workers”.

Line 119, replace “the shortest” with “shorter”.

Lines 119-120, delete “, and 119 ready to emerge as adult”.

Line 119, use past tense in method and result section. For example, “The queen bees had the shorter development……”. Please, make the edits throughout the manuscript.

Line 120-121, suggest to combine two sentences in to one “After three days, the queen eggs hatched into one-day old larvae on the fourth day……” Suggest to replace “clear” with “whitish RJ” or “transparent RJ”.

Line 121, suggest to use plural instead of plural and make the edits throughout the manuscript. For example, use “eggs” instead of egg since authors describe many egg individuals.

Line 223, suggest to delete “Africanized honey bees and”.

Line 128, replace “birth” with “hatch” and “cleans up” with “consumed all”

Line 124, change 6th to “6th”, and change the similar grammatical error across the manuscript.

Line 127, rewrite sentence, and suggested the sentence “followed by eye color changing from white to pink and then to black”.

Line 135, change “A. c. cerana queen bees” to “An A. c. cerana queen bee”

Line 139, change “birth” to “hatch” or another appropriate word throughout the paper.

Line 140, change queen bee to “queen bees”.

Line 143, add “afterwards” after “remained immobile”.

Line 144, replace “the pupa’s” with “pupae’s body”, and this suggestion is mentioned above.

Line 150, replace with “foraging for” with “ate”. “Foraging” means adult bees flight outside the bee hives to collect pollen or nectar. Newly emerged bees do not forage, instead consume stored food in hive.

Line 157, please clarify how many eggs, larvae or pupae are used for average calculation in Table 1.

Line 223, add species information “(a hybrid of the western honey bee Apis mellifera and African honey bee A. m. scutellate)” after “Africanized honey bees”.

Line 225, add “in regular honeycomb” after “worker bees”.

Line 229, delete “significant”.

Line 240, replace reach with “reached”. Use the day information consistently, and choose one between 9th day or ninth day.

Line 241, grammar error, and correct “was” to “were”.

Line 253, please add information of total amino acid content in Table 3 to support your argument.

Lines 273- 274, rewrite the sentence and it has many logic and grammatical errors.

Line 276, replace “insects” with “honey bees and other bee pollinators”.

Line 280-281, how this research guides the beekeeping practice. If authors cannot build up their relationships, please delete the sentence.

Comments on the Quality of English Language

Comments on language are included in the same file for comments and suggestions for authors.

Reviewer 3 Report

Comments and Suggestions for Authors

Manuscript ID: insects-3202758
The manuscript entitled ‘The life history and dietary diversity of queen and worker honey bees, Apis cerana cerana’’ reports interesting original research. So far, there is not much data about the biology of Apis cerana. The authors clearly present the aim of the study. The introduction summarises the current state of the topic. The methods and results are also clearly presented. Undoubtedly, the manuscript broadens our knowledge concerning the biology of Apis cerana. However, it needs a major revision.

General comments

I suggest checking the manuscript by the native speaker.

I suggest using ‘’queens’’ and ‘’workers’’ instead of ‘’queen bees’’ and ‘’worker bees’’ in the manuscript.

I also suggest adding information about which species or subspecies of bees you write in the whole manuscript. Now, it is not always clear.

Detailed comments

Intorduction

Line 24: Why specifically Apis mellifera ligustica and not Apis mellifera in general? Please explain.

Line26: What does it mean temporarily? Please specify.

Lines 33-34: ,,… developmental space…’’ What does it exactly mean? Size of comb cells? Please specify.

Lines 37-42: Is this part of the manuscript concerning A. mellifera? There are more articles showing differences in royal and worker jelly.

Lines 42-45: I suggest developing this issue a little bit. It is very important issue in the case of this manuscript.

Materials and Methods

I suggest adding more information about honey bee colonies, f.ex. size, health, the presence of pathogens or parasites, nests, food sources.

Lines 79-80: How many queens were reared? If you weighted 30 of them each day, you needed a lot of queens. All these queens were reared simultaneously or in sequence? Did you weigh different queens and workers every day or the same individuals? More details are needed.

Lines 86-88: I suggest adding more details about used methods.

Line 87: …Kjeldahl method… Please add a citation.

Statistical analyses: Did you check if there are differences among honey bee colonies? In the case of A. mellifera, there are often significant differences even among similar colonies.

Results

Lines 121-134: It is worth referring to Table 2.

Lines 129-130: As a result, a larval stage of queens lasts 6 days, and a pupal stage lasts 7 days. I suggest adding this information.

When were cells cupped?

Lines 140-141: It is worth referring to Table 2.

When were cells cupped?

Lines 148-149: …the pupa’s moisture content decrease leading to reduction in size and weight… The pupa also don't eat, whereas morphogenesis is a costly process. Was the water content in the body checked?

Line 150: As a result, a pupal stage of workers lasts 11 days. I suggest adding this information.

Caption of Fig 3. ‘’…A.c.c. worker bees in a hive…’’ in worker cells, not a hive, as a hive is a box housed by bees.

Line 156: ‘’… The average weight of an egg is 0.18 mg...’’ both queens and workers

Lines 157-158: ‘’… the average weight of a 1-day-old 157 larva being 0.54 mg....’’ both queens and workers

Table 2.

It would be interesting to see how body weight increased over time as a percentage.

The minimum and maximum of body weight is also interesting to see. Because body size and body mass of queens A. mellifera affect their abilities to lay eggs. Moreover, smaller and lighter queens are intermediate between a queen and a worker. Did you find any intercastes? Were there any differences in body weight among colonies?

I suggest adding a number of samples to Table 2.

Caption of Fig 4. A lack of one asterisk …p<0.01.

Table 3. I suggest adding full names of amino acids.

Discussion

It would be helpful to know which species or subspecies of bees are discussed when you are citing literature.

Line 225: worker cells, not a hive

Line 227: A weight of eggs? Please specify.

Lines 228-229: There are some articles about the origins and evolution of bees, different species, subspecies or lines. It would be interesting if you placed the results of this study in such a context. Why does A.c.c. show similarities to Africanized honey bees? Please explain.

Line 231: A weight of eggs?

Line  233: A weight of eggs or a weight of newly emerged queens? Please specify.

Lines 234-236: What specific is in A.c.c. ecological niche that could be responsible for such results? Explain please.

Lines 240-242: Body size and body mass also depend on the diameter of worker cells in the combs. Did you measure in this study or find in the literature such information in the case of A.c.c.?

Caste differentiation (queens or workers) in A. mellifera strongly depends on the quality and quantity of the food that larvae eat. The first few days are especially important. I suggest referring your results to this issue, including your new data about changes in queen and worker jelly.

Line 273. Lack of dot.

Supplementary

Table S1

Adding the numbers of samples would be very helpful.

Were there any differences among colonies?

‘’Type and main wt.’’ wt - what does it mean?

Round 2

Reviewer 1 Report

Comments and Suggestions for Authors

Genaral comments

Authors have made excellent efforts in revising this article. However, the article still needs some corrections. English language (grammar) at some places still needs corrections. Kindly write the whole article in Past Tense. I suggest the authors to read the article carefully and make these corrections. Therefore, the article would need a minor revision.

Specific comments

Title: OK

Abstract:

1.     Line 25: Add here 3-4 lines about the methods used for this study.

Introduction:

1.     Line 34: The A. cerana—The Chinese honey bee, A cerana

2.     Lines 36-: Here, you have evaded replies to my queries on frequent absconding and occurrence of Thai Sac Brood Disease in Apis cerana. At least you should make a mention of these in the Southeast and South-Asian Apis cerana and their absence in Chinese Apis cerana.

3.     Line 46: A. m. ligustica-Write in italics e,g. A. m. ligustica

4.     Lines 45-47: Little confusing; kindly rephrase.

5.     Lines 64-70: These are results. Delete.

Material and Methods

1.     Write this section in Past Tense.

Results :

Kindly write this section in Past Tense

Discussion: OK

Conclusion:

Lines 316-317: Rephrase as- The earlier studies on A. m. ligustica and the Africanized honey bee have provided detailed ontogenesis of queen and workers [7, 20, 22, 33, 34]. These

Rest is OK

Comments on the Quality of English Language

Needs minor editing

Reviewer 3 Report

Comments and Suggestions for Authors

There is still lack of explanation why A.c.c. was compared to specifically Apis mellifera ligustica. The western honey bee (Apis mellifera) includes other subspecies f.ex. A. m. carnica, A. m. mellifera.

Some explanations appeared in authors’ response but not in the manuscript (f.ex. developmental space line 49).

Line 253 very small sample, limitations related to such small sample should be discussed.
